# Movement and Home Range of Amur Soft-Shell Turtle (*Pelodiscus maackii*) in the Ussuri River, Heilongjiang Province, China

**DOI:** 10.3390/ani14071088

**Published:** 2024-04-03

**Authors:** Xiaochen Hou, Haitao Shi

**Affiliations:** Ministry of Education Key Laboratory for Ecology of Tropical Islands, College of Life Sciences, Hainan Normal University, Haikou 571158, China; 20211071300001@hainnu.edu.cn

**Keywords:** Amur soft-shell turtle, *Pelodiscus maackii*, spatial ecology, movement rate, home range, conservation

## Abstract

**Simple Summary:**

In our research, we investigated the home range of the Amur soft-shell turtle (*Pelodiscus maacki*i) along the Ussuri River, Heilongjiang Province, China, using radiotelemetry. Our findings revealed that juvenile turtles heavily relied on still water channels and undisturbed vegetation-rich shorelines for their survival, with their home range predominantly restricted to these environments. Interestingly, adults exhibited a broader distribution range, with greater daily distances traveled compared to juveniles, suggesting potential habitat segregation. This study offers valuable insights to inform future conservation strategies and fills a gap in knowledge of the natural history of this endangered species.

**Abstract:**

Comprehensively understanding the spatial ecology and habitat preferences of endangered species is essential for population restoration and conservation. We investigated the home range and movement of the endangered Amur soft-shell turtle (*Pelodiscus maackii*) in the Ussuri River, Heilongjiang Province, Northeastern China. The study involved tracking 19 Amur soft-shell turtles from late June to mid-October, 2022, resulting in complete and partial home range size data for eight subadults and two adults, respectively. The primary analysis focused on eight subadults, and the models that best described daily movement were identified. We also explored the potential factors influencing home range size. The mean movement rate ranged from 39.18 ± 20.04 m/day to 72.45 ± 29.36 m/day and was positively correlated with the linear home range and water temperature. The most enlightening estimation of home range was derived from a 95% kernel density estimate, utilizing likelihood cross-validation smoothing while adhering to constraints delineated by the river boundaries. The average size of the home range was determined to be 1.02 hectares and displayed no correlation with body size. Subadults tended to establish well-defined home ranges over time, whereas defining home ranges for adults proved challenging. This research addresses a gap regarding the ecology of the Amur soft-shell turtle and provides a foundation for future conservation plans.

## 1. Introduction

The Amur Soft-shell turtle (*Pelodiscus maackii,* hereafter AST) [1] is a freshwater turtle species that has historically thrived across Northeast China, the Russian Far East, the Korean Peninsula, and Japan [1,2]. However, the population in Northeastern China has drastically declined in recent decades, owing to the effects of anthropogenic activities, such as uncontrolled hunting and habitat destruction, and the species has completely disappeared from much of its original range (Hou. Unpublished content). Although described as early as the 19th century and likely currently critically endangered, research on this species is desperately lacking. Only a few articles, i.e., Baek et al. [3] on the sequencing of its complete mitochondrial genome and Chang et al. [4] and Suzuki and Hikida [5] on surveys of freshwater turtle species in South Korea and Japan, have morphologically and molecularly proved that ASTs inhabit the rivers of both countries and are suspected to be native to both regions. Overall, there is a notable absence of fundamental ecological research on ASTs throughout their distribution range, which presents a barrier to comprehending their environmental needs in natural habitats and hinders endeavors to effectively evaluate and protect this species.

The significance of spatial ecology in riverine turtles lies in its direct linkage to their adaptations for resource acquisition, reproduction, and survival within the constantly fluctuating dynamics of their environment [6]. Recognizing the critical role of spatial data in unraveling the ecology of an organism and guiding protective management, this study aimed to examine the movement and home ranges of *P. maackii.* The current study is important because it fills a knowledge gap regarding the natural history of this poorly studied species and may establish a theoretical foundation for future conservation efforts. Another objective of this study was to provide a set of management recommendations for the protection of ASTs in the Ussuri River.

## 2. Materials and Methods

### 2.1. Study Site, Radio Telemetry, and Data Collection

The study was conducted near Jewelry Island, situated mid-stream in the Ussuri River, Heilongjiang Province, China (46°44′–46°49′ N, 133°83′–133°85′ E) (Figure 1). The study area lies to the south of Jewelry Island, encompassing two primary water channels and a segment of a tributary known as Small Wood Creek. Within this region, shallow water bodies with depths < 1.5 m are primarily situated near the shoreline and water channel 2. Water channel 1 and Small Wood Creek are deeper, with depths ranging from 2.5 to 3 m. The main watercourse of the Ussuri River maintains depths of 4–6 m. The bottom sediment in the area is predominantly of muddy and sandy composition, with sandy soil prevailing along the main watercourse shoreline and sandy gravel content in the upper reaches of Small Wood Creek. The vegetation along the shoreline is dominated by willow (*Salix* sp.), reed (*Phragmites* sp.), and sedge (*Cyperus* sp.). The climate of this area is cold temperate continental monsoon, characterized by four distinct seasons with a long cold winter and short warm summer. The average annual temperature is 3.5 °C, with an extreme low temperature of −36.1 °C recorded in winter and an extreme high temperature of 35.2 °C recorded in summer. Most of the study area is relatively undisturbed, with only a small portion featuring an altered shoreline formed by piles of large boulders in the southern section. The human population density in this region is minimal, although it is frequently visited by fishermen that fish for a living. A fishing settlement is located in the northern part of the research area, near the mouth of Small Wood Creek. Because the Ussuri River serves as a border between Russia and China, our study was confined to the Chinese side of the river, precluding the possibility of exploration on the Russian side.

Specially designed traps (Figure 2) were deployed in this region to capture ASTs, and fish meat and guts were used as bait. Traps were checked and rebaited daily. However, because of the overall scarcity of ASTs in the Ussuri River (Chinese side), we did not catch a sufficient number of suitably sized specimens for our telemetry research. Therefore, we enlisted local fishers to assist in capturing ASTs to supplement our samples (Figure 3). When a novel specimen was obtained, we measured body mass with precision to the nearest gram with electronic scales; plastron length (PL) was measured to the nearest millimeter using tree calipers. Sex was distinguished on the basis of multiple characteristics similar to those of other soft-shell turtle species. To the best of our knowledge, there is no literature available on the means by which to determine the size at sexual maturity of ASTs. However, according to several observations of fishermen processing captured ASTs, we found that female Amur soft-shell turtles weighing around 900 g had begun to display mature ovaries, whereas males around 800 g had basically fully developed secondary sexual characteristics. Hence, we used these criteria to determine the sexual maturity of ASTs.

We attached high-frequency transmitters (164 MHz; AL-2F, Holohil Systems Ltd., Caro, ON, Canada) onto the turtles by drilling holes in the posterior side of the carapace with a handheld electric drill with 1.8 mm caliber of the drill bit and then fastening the transmitters in place using 2.60 mm wide zip ties threaded through the drilled holes. Soft white rubber cushions were applied beneath the transmitter, safeguarding the turtle’s skin and the corresponding ventral area. The ties were fixed to these cushions, ensuring no direct contact with the turtle’s soft connective tissue, thereby averting any potential abrasions or harm to the ASTs (Figure 4). Following the transmitter’s secure attachment, the wound was disinfected using iodine. Subsequently, the ASTs were carefully placed in a dry environment overnight to mitigate the risk of wound infection. The radio transmitter weight did not exceed 8% of the body mass of the turtle. A TRX-1000S receiver (Carbondale, IL, USA) and three-element antenna were used to track the radio-tagged turtles. After releasing the turtles into the study region, they were allowed to acclimatize for 1 week before commencing the tracking procedure. Turtles were intensively tracked daily by boat and occasionally by wading from June, 2022, to mid-October, 2022, coinciding with the peak activity periods of ASTs in this region (Hou. Unpublished content). The telemetry schedule was adjusted according to weather conditions, typically operating between 7:00 a.m. and 5:00 p.m. One location was telemetried each day for each ASTs. The position data were recorded via a handheld global positioning system (GPS map 78s, Garmin International Inc., Olathe, KS, USA). On occasions where a close approach was not attainable, such as when the turtles ventured into the Russian side, locations were determined by triangulation from the nearest accessible areas to the turtles. We measured the water temperature at 12:00 noon daily at a fixed position with water depth of 50 cm, using a substrate thermometer as a predictor variable for daily movement rates. Upon finishing the tracking study, all radio transmitters were carefully detached from the turtles, and all turtles were released in good health. No instances of infections or health complications arising from the transmitter’s attachment were noted. Our research was approved by the Fisheries Department of Hulin County.

### 2.2. Home Range Analysis

To study the spatial utilization of each radio-tracked AST, we calculated the linear range, 100% minimum convex polygon (MCP), and fixed kernel density estimates (KDEs) at 50% and 95% levels. The linear range, as defined by Sexton [8], represents the direct distance between the two furthest locations and serves as an indicator of turtle activity intensity and capability over a given time period [9]. While the MCP method may offer graphical simplicity and exhibit reasonable stability for comparisons [10], it is important to note that this approach may include extensive areas that are never used by animals, as indicated by previous studies [11,12]. Kernel analysis uses a nonparametric probability density approach to estimate the home range size [12,13], which has the potential to reveal variations in the intensity of usage within the home range. A crucial aspect of kernel analysis is the selection of an appropriate value for the smoothing parameter h [14], which is predominantly addressed by two widely adopted methods: least squares cross-validation (LSCV) and likelihood cross-validation (CVh) [13,14,15]. However, because of its perceived drawbacks and inferior performance [16,17,18,19], LSCV was generally discounted, and we selected CVh to calculate the smoothing parameter in our analysis. Smoothing parameters were calculated using the Animal Space Use Program [20]. We followed the guidelines provided by Row and Blouin-Demers [21], who advocated adjusting the smoothing parameter to achieve equivalence as much as possible between the 95% kernel area and the MCP area in home range size studies of herpetofauna to achieve a more accurate analysis. Hence, for comparison, we chose the calculated parameter that aligned 95% of the kernel area most closely with the corresponding MCP area for all turtles. We calculated the MCP and kernel density estimates using the minimum bounding geometry and kernel density tools in ArcGIS Pro 3.2 [7]. For animals inhabiting linear habitats, area estimates often encompass substantial areas of unutilized space [22]; hence, we clipped the MCP and kernel density contours to the river [23]. Linear range (LR) size was calculated in ArcGIS pro [7], and as the species predominantly inhabits aquatic environments, LRs traversing terrestrial areas were adjusted to reflect the shortest distance through water [24]. The degree of home range overlap for each turtle was quantified as the proportion of its entire home range that intersected with the ranges of other turtles [25]. The analysis was performed using the R package ‘adehabitat’ [26]. The relationship between range size and body size was determined using Spearman’s rank correlation test. Significance was determined at α = 0.05.

On several occasions, radio transmitters were displaced from turtles by tangling to fish nets or twigs after a period of radio tracking, and we used bootstrapped MCP to determine whether the home range size estimates for these turtles approached an asymptote with the acquired location data [27]. For each turtle, we employed a bootstrapping method to estimate the MCP home range area by incrementally adding radiolocations, with 100 iterations per increment, using the R package ‘Move’ [28]. To determine if the home range size estimates became asymptotic, we used nonlinear regression with a monomolecular growth function [29] to estimate the asymptotic home range size. If the MCP home range size estimate was ≥90% of the asymptote, this suggests that we had gathered a satisfactory number of radiolocations to precisely establish the individual’s home range, and the home range estimate was deemed complete. Turtles exhibiting non-asymptotic MCP home range estimates were disregarded for subsequent analyses.

### 2.3. Movement Analysis

Considering the predominant aquatic behavior of ASTs, we operated under the assumption that their movements were restricted to the river channel. We determined the movement distance (m) through the river using the least-cost path tools in ArcGIS Pro [7], which calculated the shortest path between consecutive points along a polygon representing the river. For points where the distances traversed land areas, we used polygon-to-raster tools in ArcGIS Pro [7] to compute the shortest curved pathways between points through the river channels. This method illustrates the minimal movement necessary to travel between these locations within river channels. Although the turtles may have taken more extended routes, quantifying such possibilities remains impossible. Therefore, we based our comparisons of movement rates on the minimum distance traveled between locations.

We calculated the daily movement rate in meters/day by dividing the distance between successive locations by the number of days. Subsequently, we computed the minimum, maximum, and mean movement rates for each turtle. To explore potential factors influencing the movement rate, we utilized a mixed-effects model. Predictor variables were chosen based on previous studies and our experience and included water temperature [30], sex [31], LR, and PL. All continuous predictor variables underwent z-transformation, standardizing them into standard deviation (SD) units. Seventeen potential mixed-effects models were formulated to describe the movement rate (m/d) (full model structures are provided in Appendix B Table A2). To evaluate multicollinearity among the predictor variables, we calculated the variance inflation factor (VIF) utilizing the ‘vif’ function from the R package ‘usdm’ [32]. A VIF >10 signified the presence of problematic multicollinearity [33] and required a reexamination of the variables. The null model consisted solely of the intercept, while the global model included all covariates and two-way interactions. We assessed the residual plot and histogram of the global model for patterns and normality. In instances where the assumptions of normality and variance posed challenges, we opted for a natural log transformation of the movement rate variable. Additionally, quadratic terms for water temperature were introduced to account for potential fluctuations in movement patterns, allowing for peaks and troughs to be accommodated. Individual turtles were included in the random effects analysis. When conducting the analyses, we only included turtles with asymptotic home range size estimates because we used the complete LR as a predictor variable. Mixed-effects models were constructed using R version 4.2.0 [34], employing the R package ‘lme4’ [35]. The model selection was conducted based on AICC scores utilizing the R package ‘AICcmodavg’ [36]. Models within two AICC units of the top model were considered to have substantial support and were included in their respective confidence sets [37]. A parameter estimation for the top model was performed using restricted maximum likelihood. Predictors were deemed strong if their 95% confidence intervals (CIs) did not significantly overlap zero. Marginal and conditional R^2^ values were computed using the ‘r.squaredGLMM’ function from the R package ‘MuMIn’ [38,39].

## 3. Results

We attached radio transmitters to 12 subadults (seven females and five males, weighing 520–680 g) and seven adults (three males and four females, weighing 910–2950 g). The mean PL for subadults was 13.9 ± 0.43 (SD) cm and that for adults was 22.11 ± 4.30 (SD) cm (Table A1). We acquired a complete home range estimates for 8 of the 12 subadults; however, no complete home range estimates were obtained for any adult (Figure 5, Table A1). Subadults weighing <700 g tended to primarily inhabit the river’s edge, restricting their movement to the vegetated shoreline and slow-velocity water channels. They rarely ventured into deep open water for unknown reasons and easily established a home range within our study area. Unfortunately, our study was greatly influenced by intense fishing activities in this area, leading to a notable effect as evidenced by the entanglement of some specimens in fish nets, causing the detachment of radio transmitters. In contrast to subadult ASTs, adults, particularly those of a large size, exhibited a tendency to explore deeper water areas. Unlike younger turtles, mature turtles never confined their active range to the shoreline and water channels. Compelling evidence suggested that adult turtles, particularly larger ones, confidently utilized both sides of the river, extending their home range well into the Russian territory, which was inaccessible during this study due to permit restrictions. In addition, the telemetry study showed that large adults could rapidly move large distances. On several occasions, signals from adult ASTs vanished within the second or third day post release (Table A1) despite comprehensive searches conducted several kilometers upstream and downstream. Limitations in equipment and personnel constrained us from regularly extending the search beyond 10 km, and we were unable to cross the middle line of the river into the Russian territory. Hence, determining whether these adults migrated to the waterways on the Russian side, or traveled an extensive long distance upstream or downstream, remains challenging. A total of 672 locations were recorded between 20 June and 10 October 2022 for all turtles. No mortality was observed during the study period.

### 3.1. Home Range Analysis

Clipped KDEs, MCP, and LR were calculated for eight subadult turtles for which the complete home range size estimates were achieved, along with two adult turtles (ID 13 and 14) with >10 locations for comparison (Figure 6). The chosen calculated CVh smoothing parameter was 17 for eight subadults and 120 for two adults. The number of core areas (clipped 50% KDE CVh) varied from one to five for all turtles.

The mean LR size was 693 ± 238 (SD) m and ranged from 388 to 1145 m among the eight subadults, with a coefficient of variation (CV) of 34% (Table A1; Figure 6). There was no statistically significant relationship between the LR and either PL or body mass (BM) (LR and PL: *r_s_* = −4.52, *p* = 0.27, n = 8; LR and BM: *r_s_* = −0.429, *p* = 0.27, n = 8). The sizes of MCPs among eight subadults ranged from 1.30 to 3.65 ha (CV = 35%) with a mean of 2.03 ± 0.72 (SD) ha (Table A1; Figure 6). There was no significant relationship between MCP size and body size (MCP and PL: *r_s_* = −0.286, *p* = 0.5, n = 8; MCP and BM: *r_s_* = −0.262, *p* = 0.54, n = 8). Moreover, 95% and 50% KDEs ranged from 0.66 to 2.07 ha (CV = 43%) and 0.07 to 0.18 ha (CV = 34%), respectively, with mean sizes of 1.02 ± 0.44 (SD) ha and 0.099 ± 0.034 (SD), respectively (Table A1; Figure 6). No significant relationships between the KDE and body size for eight subadults turtles were observed (95% KDE and PL: *r_s_* = −0.443, *p* = 0.27, n = 8; 95% KDE and BM: *r_s_* = −0.335, *p* = 0.42, n = 8; 50% KDE and PL: *r_s_* = −0.449, *p* = 0.27, n = 8; 50% KDE and BM: *r_s_* = −0.424, *p* = 0.29, n = 8).

The MCP of each turtle overlapped with the MCPs of 3–7 other turtles (mean = 0.91 ± 0.57 ha SD, n = 8). Mean MCP overlap areas varied from 0.65 ± 0.43 (SD) ha to 1.73 ± 0.45 (SD) ha. The total 95% KDEs overlapped with 3–7 other turtles (mean = 0.48 ± 0.23 [SD] ha, n = 8), with mean areas of overlap ranging from 0.28 ± 0.10 (SD) ha to 0.77 ± 0.33 (SD) ha. The 50% KDEs overlapped with 2–7 core areas of other turtles (mean = 0.047 ± 0.021 [SD] ha [n = 8]), with mean areas of overlap ranging from 0.024 ± 0.008 (SD) ha to 0.078 ± 0.008 (SD) ha (Table 1).

### 3.2. Movement Analysis

We used eight subadult turtles with complete home range size estimates for movement analysis. The turtles exhibited infrequent extended stays at single locations. The count of movements recorded for each turtle ranged from 57 to 78, minimum movement rates varied from 8.5 to 18.6 m/day for all turtles, and maximum movement rates varied from 89.9 to 275.6 m/day for all turtles (Table A1). Furthermore, telemetry studies of limited locations acquired from individuals 13 and 14 revealed that the daily distances traveled by adults were significantly greater than those travelled by immature ASTs (Table A1). However, we did not obtain sufficient data to analyze this phenomenon in a systematic and in-depth manner.

All predictor variables in our movement rate models exhibited VIF values of <3.2, suggesting that multicollinearity was not problematic. To normalize the data, we employed a natural log transformation of movement rates (back-transformed after analyzing the figures).

The top mixed effects model for the movement rate was the “global model” (Table 2 and Table A2). Both the LR and water temperature showed confidence intervals that did not overlap with zero (Table 3). The movement rate was positively correlated with the LR and water temperature (Figure 7). The rest terms showed less pronounced effects, with confidence intervals largely overlapping zero (Table 3).

## 4. Discussion

### 4.1. Home Range

The rationale behind selecting this study area stemmed from catch records supplied by local fishermen. These records revealed that turtles were sporadically encountered at this specific location, whereas turtles were seldom or never captured in other adjacent river sections. This information led us to infer that this particular area is the preferred habitat for ASTs, rendering it an ideal location for ecological studies.

Among the eight subadult ASTs for which we estimated a complete home range size in the study area, the individual LRs exhibited considerable variability, ranging from 388 to 1145 m. This notable divergence in LRs between individuals has also been observed in other soft-shell turtle species [40,41]. The MCP estimator yields values ranged from 1.30 to 3.65 ha. It is worth noting that this method may encompass areas that animals have never used, potentially resulting in an overestimation of the actual range size. Despite this limitation, the MCP is widely employed for home range estimates and is commonly used to enable inter- and intra-species comparisons across various studies [42,43]. The KDEs also encompassed areas not utilized by turtles but to a much lesser extent. Clipping the KDE to the river boundary helped define a more accurate representation of space utilization [23]. Notably, KDEs using CVh smoothing yielded larger estimates than those using LSCV smoothing. LSCVs tend to be under-smoothed [19], which may result in unrealistically fragmented home ranges [44]. The ideal smoothing parameter lacks consensus, and the under-smoothing observed with LSCV fails to delineate reasonable activity areas for this highly mobile species. We advocate for employing clipped 95% KDEs with CVh smoothing as it furnishes a meaningful estimation of home range for ASTs, with the clipped 50% KDE (CVh) accurately depicting core areas. [43]. Some AST home ranges included a few core areas, whereas those of others included several. This variation is not surprising considering the differences observed in home range sizes. It is widely accepted that animals often exhibit multimodal home ranges [45], and space use often varies along a continuum from sedentary to nomadic [46]. In general, turtles employ diverse foraging tactics, likely influenced by balancing trade-offs in utilizing various resources [47], leading to variations in home ranges among individuals.

Serial autocorrelation, wherein radiolocations exhibit partial dependence on previous locations, has the potential to compromise the validity of home range estimates [48]. This effect was most pronounced when locations were recorded at very short time intervals, restricting the animals from moving to different areas. In our study, we mitigated autocorrelation concerns as radiotelemetry locations were never taken on the same day, and the ASTs demonstrated the capability of traversing large distances in a short time. To define a complete home range with sufficient samples, the home range size estimate must reach an asymptotic value for animals that exhibit limited wandering or nomadic behavior [27,49]. Home range estimates for subadult ASTs that were never interrupted or were interrupted by fishing activity after an extensive radio tracking period would be asymptotic with a well-defined home range, indicating that immature ASTs did not behave nomadically. This is likely because of the tendency of subadult ASTs to restrict their range to specific areas near the shoreline with abundant vegetation and calm shallow water bodies.

There was a certain amount of home range overlap among subadult ASTs. Spearman’s rank correlation analysis revealed no association between home range size and body size. This lack of correlation may have been attributed to the fact that the analyzed individuals were in approximately the same weight range with small variations, making it challenging to discern a meaningful relationship between home range size and body size. Nevertheless, body size may be an important factor when considering large numbers of individuals.

### 4.2. Movement

The extent of river turtle movement fluctuates according to the specific life history demands associated with foraging, reproduction, and survival [6]. The active period for ASTs is from mid-May to mid-October (Hou, Unpublished content); therefore, our study covered the most active period. Turtles were rarely inactive during the tracking period, and adult and subadult ASTs differed significantly in their daily movement patterns. Although the location data for adult ASTs were limited, the available information indicated that both the mean and maximum daily movement rates of subadult ASTs were considerably lower than those of their adult counterparts. The reason for the shorter daily movement rate in subadults may have been based on the following three points. First, these subadults appeared reluctant to enter open deep water; they mainly confined their activity to relatively limited areas surrounded by deep water, rendering frequent and extensive movement unnecessary. The size of the water body is thought to be a crucial factor influencing the scale of movement exhibited by riverine organisms [50]. Moreover, variations in resource distribution can lead to diverse movement patterns with densely concentrated resources contributing to reduced movement rates [46]. These vegetated shorelines and slow-velocity water channels provide a habitat rich in small organisms such as insect larvae, crustaceans, mollusks, snails, amphibians, and small fish [51], which are potential prey for juvenile ASTs, diminishing the necessity to cover extensive distances in the search for food. Another possible explanation is that immature ASTs are not subjected to breeding pressures, unlike adults who must cover greater distances to search for mates or suitable egg-laying sites [52]. Consequently, this lack of breeding-induced impulsion may lead immature ASTs to adopt a distinct movement pattern compared to that of adults.

The telemetry data acquired for adult ASTs indicated that they did not restrict their range to shallow, nearshore areas with abundant vegetation or to water channels. The daily movement rates of adult ASTs observed in our study were similar to those documented for adult smooth soft-shell turtles (*Apalone mutica*) in the Kaskaskia River in Illinois, North America [41]. However, this previous study did not include subadults; therefore, we could not make a relevant comparison. Some sudden loss of signals from several larger adults within a brief period is likely attributable to an escape response triggered by the stress induced by capture and relocation [40]. Notably, one released adult female (2720 g) was recaptured the following spring 55 km upstream with the radio transmitter still attached; this distance was beyond our tracking capabilities.

The mixed-effects model showed a significant positive correlation between water temperature and the daily movement rate. We observed no discernible peak in the movement rate at intermediate temperatures, which was followed by a decline as the temperature increased further [41,53,54]. This may have been attributed to the relatively low water temperatures in the study area, which is characterized by a temperate climate. The maximum water temperature in summer in our study area does not exceed 26 ℃, which does not impose thermal inhibition on the activity level of ASTs. Consequently, it is reasonable that the movement rate of ASTs inhabiting this region exhibited a monotonic positive correlation with water temperature without experiencing estivation, owing to excessively high water temperatures, as observed in other freshwater turtle species from warmer climates [55]. Another factor showing a positive correlation with the daily movement rate was LR, which serves as an indicator of the active range of animals [9]. Notably, individuals with an LR of 1145 m also had the highest average daily movement rate. This correlation indicated that ASTs with broader activity ranges also demonstrated greater mobility. Such varied movement patterns among animals residing in the same habitat have shown the potential for diverse strategies for resource exploration among different individuals [56,57].

### 4.3. Conservation and Management

Based on our investigation, we found a lack of awareness regarding the uniqueness and rarity of ASTs among local residents, who often mistake ASTs for common consumable Chinese soft-shelled turtles (*Pelodiscus sinensis*). Unfortunately, this misconception results in the total absence of protective measures by local authorities. Consequently, there are no constraints on hunting and selling ASTs, which further exacerbates the decline in this endangered species. Relevant protection measures must be implemented immediately to avoid further population decline. Moreover, these vital shallowly vegetated shorelines and undisturbed water channels play crucial roles in the survival of immature ASTs. Unfortunately, these areas have also experienced more intense fishing activities, which has had a severe negative effect on the immature turtles residing there. The recurrent use of fishing nets in these critical zones resulted in the entanglement of several subadult specimens during the study. Frequent fishing activities in these essential habitats can have a substantial detrimental effect on the long-term survival of ASTs in the Ussuri River.

Because of the intense fishing activities along the Chinese stretch of the Ussuri River, the AST population has suffered severe setbacks. We speculated that the persistence of the AST population in the Ussuri River was primarily because the opposite Russian territory remained virtually untouched and free from human interference or fishing activities, creating a crucial sanctuary for the sustenance of the population. Despite this, the AST population in the Ussuri River remains at risk of collapse due to intense anthropogenic impacts on the Chinese side of the river. Hence, it is imperative to protect ASTs in China, particularly in areas that are critical for the survival of immature turtles.

## 5. Conclusions

Given the constraints inherent to our study, we were only able to delineate the home ranges and movement patterns of subadult ASTs. Nonetheless, the importance of the immature phase in the lifecycle of organisms cannot be overlooked, and exploring the ecological characteristics of immature individuals has significant biological value. To comprehensively understand the ecology of ASTs at various life stages, future research should focus on the home range and other ecological facets of adult ASTs. This may require a heightened commitment of labor, equipment, financial resources, and additional administration to conduct comprehensive telemetry studies spanning both Chinese and Russian territories. Notably, our findings underscore the need to preserve vital undisturbed areas for the survival of immature ASTs and urge future conservation efforts to focus on these specific habitats.

Finally, major flooding occurred at the end of our study, which forced the cessation of our investigation. The effect of extended overbank flow on the movement and home ranges of turtles remains uncertain. Whether immature ASTs would utilize such flooding events for dispersal migration remains unknown. Future research endeavors should investigate the spatial ecology of ASTs during and after significant flood occurrences to clarify the impact on movement rates and home range sizes.

## Figures and Tables

**Figure 1 animals-14-01088-f001:**
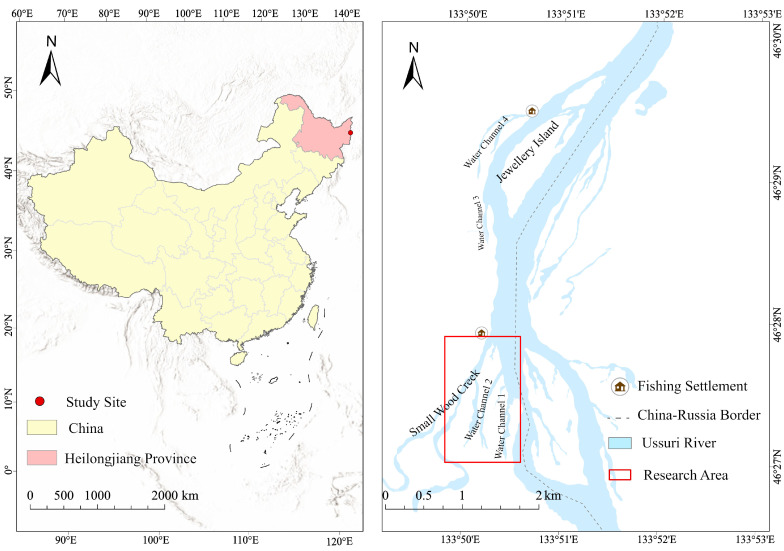
Heilongjiang Province, China. The red dot indicates the study site in the Ussuri River, and the red square denotes the research area. Maps were designed using ArcGis Pro 3.2 [7].

**Figure 2 animals-14-01088-f002:**
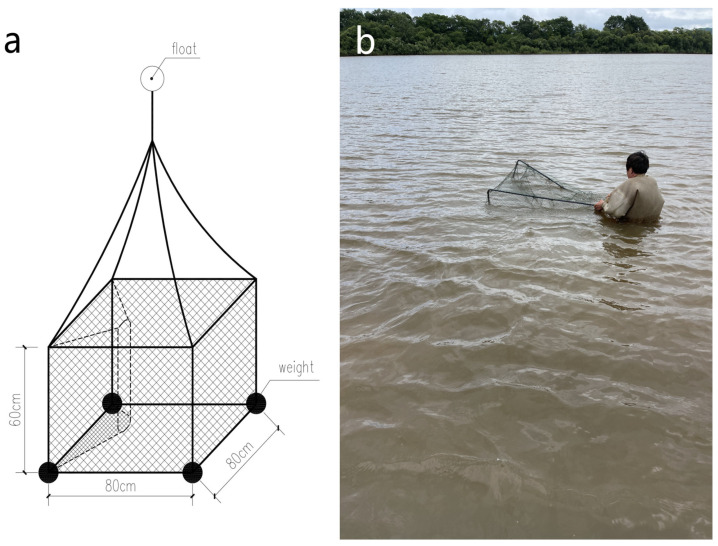
(**a**) Industrial drawing of cage trap designed. (**b**) Cage deployment.

**Figure 3 animals-14-01088-f003:**
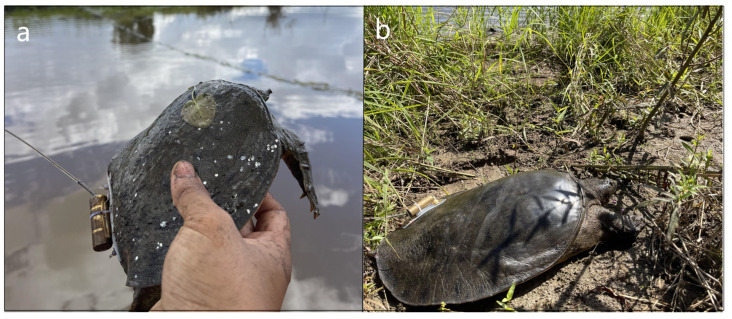
(**a**) Subadult male (590 g) and (**b**) adult female (2720 g) *Pelodiscus maackii* with radio transmitters attached in the Ussuri River, China, in 2022.

**Figure 4 animals-14-01088-f004:**
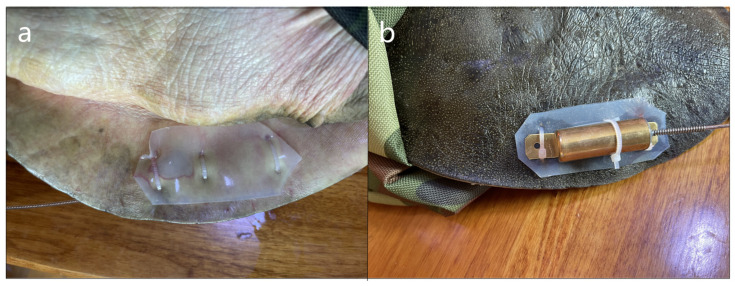
Radio transmitter attachment for *P. maackii* seen from (**a**) ventral view and (**b**) dorsal view.

**Figure 5 animals-14-01088-f005:**
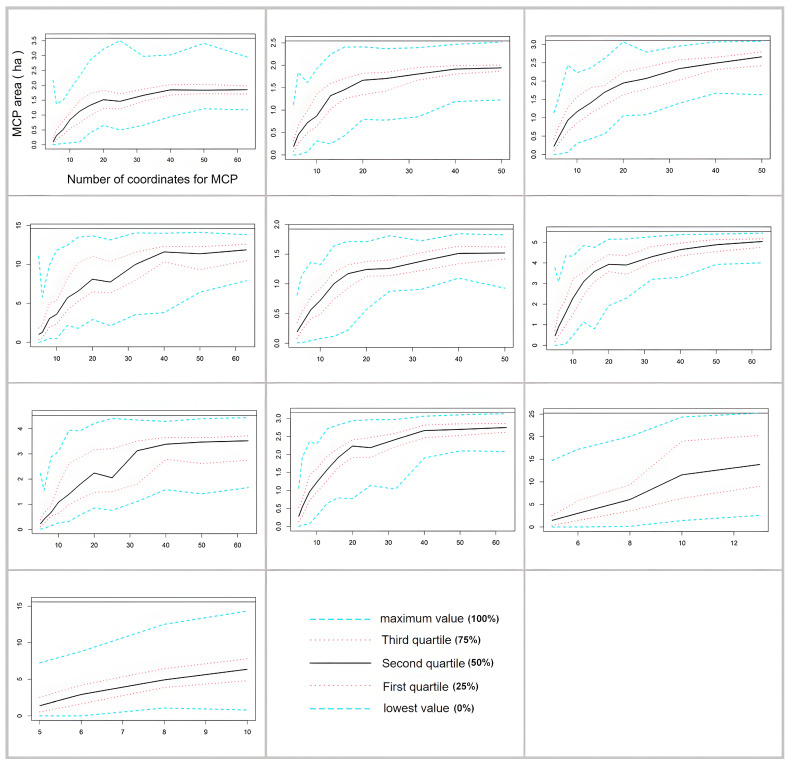
Incremental area analysis plot for 10 *P. maackii* specimens. The horizontal axis represents the number of coordinates for the minimum convex polygon (MCP) home range size estimates, and the vertical axis represents the range of the MCP area. Different lines denote minimum, quartile, and maximum values of bootstrap iterations.

**Figure 6 animals-14-01088-f006:**
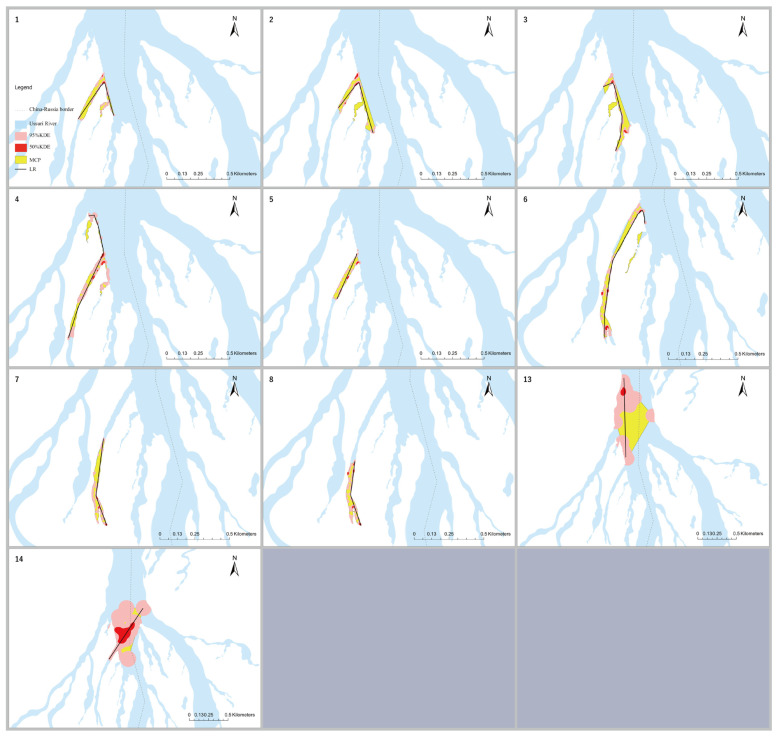
Study area showing radio tracked *P. maackii* individuals (1–8,13,14, Appendix A Table A1) MCPs, linear home ranges (LRs), 95% kernel density estimates (KDEs), and 50% KDE home range estimates. Maps were designed using ArcGIS Pro 3.2 [7].

**Figure 7 animals-14-01088-f007:**
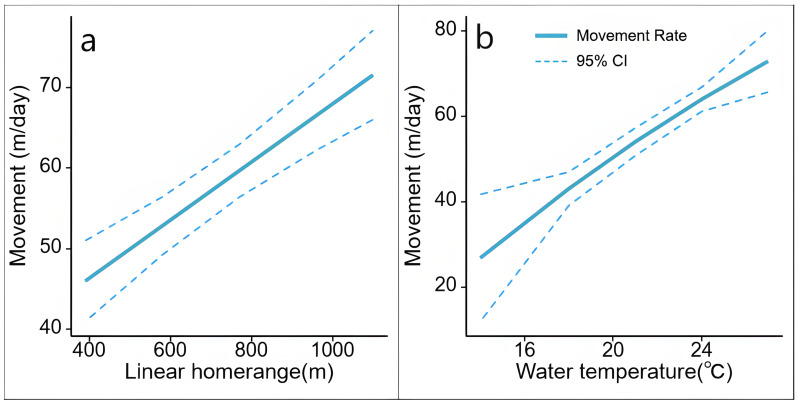
Movement rate (m/day) and 95% confidence intervals for *P. maackii* for covariate effects of (**a**) linear home range (m); (**b**) water temperature (°C).

**Table 1 animals-14-01088-t001:** Overlap of home range areas of eight subadult *P. maackii* for which we have complete home range size estimates, presented as the percentage of the total range of the individuals listed in the left column (see Table A1 for definition of abbreviations).

**MCP**								
**ID**	**1**	**2**	**3**	**4**	**5**	**6**	**7**	**8**
1		88.89	55.84	20.53	84.04	38.5	0	0
2	63.01		80.14	17.38	73.08	30.81	0	0
3	48.42	31.97		15.96	17.4	15.67	0	0
4	83.63	70.84	37.42		100	64.68	41.49	8.55
5	38.12	45.89	10.77	13.18		29.86	0	0
6	0	35.77	12.7	76.91	82.82		81.05	80.52
7	0	0	0	6.67	0	56.49		90.47
8	0	0	0	1.85	0	43.03	69.31	
**95% KDE**								
**ID**	**1**	**2**	**3**	**4**	**5**	**6**	**7**	**8**
1		64.75	54.78	36.92	68.86	29.85	0	0
2	0.73		49.79	31.49	76.58	31.14	0	0
3	0.56	44.96		20.07	30.41	22.45	0	0
4	0.78	86.75	61.26		97	43.09	24.1	19.83
5	0.65	79.9	31.87	34.4		22.8	0	0
6	0.35	42.38	0	40.31	45.56		82.19	72.53
7	0	0	0	9.21	0	54.55		86.94
8	0	0	0	7.46	0	47.11	69.66	
**50% KDE**								
**ID**	**1**	**2**	**3**	**4**	**5**	**6**	**7**	**8**
1		51.3	43.31	28.63	35.97	24.32	0	0
2	88.54		42.42	43.39	63.79	23.89	0	0
3	80.45	47.29		23.3	28.81	22.76	0	0
4	100	92.16	42.06		96.5	28.43	0	0
5	77.09	76.88	41.43	39.47		23.45	0	0
6	0	30.38	34.43	27.71	34.33		74.64	80.79
7	0	0	0	0	0	33.17		45.04
8	0	0	0	0	0	52.96	53.93	

**Table 2 animals-14-01088-t002:** Model rank, model name, number of parameters (K), −2 Log likelihood (−2LL), Akaike information criterion adjusted for small samples (AICc), difference in AICc value from the top model. value (∆AICc), Akaike weight (*w_i_*), marginal R^2^ (R^2^[*m*]), and conditional R^2^(R^2^[*c*]) for the top five and intercept-only mixed effects models describing movement rate of *P. maackii* in the Ussuri River, China (full model structures are provided in Appendix B, Table A2).

Rank	Model Name	K	−2LL	AICc	∆AICc	wi	R^2^(m)	R^2^(c)
1	Global	11	−2284.17	4590.87	0	0.83	0.26	0.26
2	Temp., linear home range, and sex interaction	9	−2288.68	4595.73	4.86	0.07	0.26	0.26
3	Main linear effect	7	−2291.24	4596.71	5.84	0.04	0.25	0.26
4	Temp. and linear home range interaction	7	−2292.5	4599.23	8.36	0.01	0.26	0.26
5	Temp., linear home range, and body size interaction	9	−2290.54	4599.45	8.57	0.01	0.26	0.26
17	Null	3	−2345.27	4696.58	105.71	0	0	0.15

**Table 3 animals-14-01088-t003:** Real parameter estimates, standard error, lower confidence interval, and upper confidence interval for the global mixed-effects model describing the movement rate of *P. maackii* in the Ussuri River, China.

Effect	Estimate	SE	LCI	UCI
Intercept	54.83	2.49	49.94	59.72
Water Temp.	12.6	1.92	8.84	16.37
(Water Temp.)^2^	−0.49	1.19	−2.81	1.84
Linear Home Range (LR)	8.71	1.45	5.87	11.55
Sex	5.5	4.51	−3.35	14.35
Plastron Length (PL)	−2.35	1.97	−6.22	1.51
Water Temp × LR	−1.73	1.28	−4.24	0.78
Water Temp × PL	2.8	2.05	−1.21	6.82
Water Temp × Sex	−7.1	4.74	−16.39	2.2

## Data Availability

Data will be made available upon request.

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
