# Peer review of "Movement and Home Range of Amur Soft-Shell Turtle (Pelodiscus maackii) in the Ussuri River, Heilongjiang Province, China"

_animals, 2024, doi:10.3390/ani14071088_

Round 1

Reviewer 1 Report

Comments and Suggestions for Authors

The ms is devoted to the important problem of the movement and home range of declined species Pelodiscus maackii in the Ussuri River, on the territory of China. The authos discussed very actual conservation issues and "speculated that the persistence of the population in the Ussuri River was primarily because the opposite Russian territory remained virtually untouched and free from human interference or fishing activities, creating a crucial sanctuary for sustenance of the population". However we can not see the references to the literature about the populations from the Russian Far East what is very important for conlclusions:

Adnagulov E. V., Maslova I. V. 2005. On the Distribution of Pelodiscus sinensis (Wiegmann, 1834) (Testudines: Trionychidae) in the Russian Far East // Ananjeva N. and Tsinenko O. (eds.), Herpetologia Petropolitana. Proceedings of 12th Ordinary Meeting of Societas Europaea Herpetologica, August 12–16, 2003. St. Petersburg. Russian Journal of Herpetology (supplement). pp. 117–119.

Turtle Taxonomy Working Group [Rhodin A.G.J., Iverson J.B., Bour R., Fritz U., Georges A., Shaffer H.B., van Dijk P.P.]. 2017. Turtles of the World: Annotated Checklist and Atlas of Taxonomy, Synonymy, Distribution, and Conservation Status (8th Ed.). In: Rhodin A.G.J., Iverson J.B., van Dijk P.P., Saumure R.A., Buhlman K.A., Pritchard P.C.H., Mittelmeier R.A. (Eds.). Conservation Biology of Freshwater Turtles and Tortoises: A Compilation Project of the IUCN/SSC Tortoise and Freshwater Turtle Specialist Group. Chelonian Research Monographs 7:1–292. doi: 10.3854/crm.7.checklist.atlas.v8.2017.

Maslova I.V. 2016. The protection of amphibians and reptiles in the Russian Far East // Nature Conservation Research. З. 1. № 3. p. 26–35.

Maslova I.V., Portnyagina E.Yu., Sokolova D.A., Vorobyeva P.A., Akulenko M.V., Portnyagin A.S., Somov A.A. 2018. Distribution of Rare and Endangered Amphibians and Reptiles in Primorsky Krai (Far East, Russia) // Nature Conservation Research. Заповедная наука. V. 3, Suppl. 1. pp. 61–72.

Author Response

Dear reviewer

Thank you very much for taking the time to review this manuscript.

Regarding the speculation on the Russian side of the Ussuri River, I've thoroughly reviewed the documents you referenced. However, the segment concerning P.maackii does not specifically address the population residing on the Russian side of the Ussuri River; rather, they predominantly surveys the distribution site of P.maackii populations in the Russian Far East. Consequently, it may not align with the context. Our inference stems from direct observations made along the Ussuri River. The disparities in fishing activity and human disturbance levels between the Chinese and Russian territories bordering the river are remarkable. So I think it's worth clarifying this point, and speculate that the reason that the survival of the P.maackii population along the Ussuri River, unlike the inland water populations of northeastern China which have faced extinction, is likely attributed to the pristine and undisturbed nature of the Russian side of the Ussuri River, where human intervention is minimal. Unfortunately, the existing literatures fails to adequately capture this distinctive scenario. I hope you can understan. Thank you very much for your review and valuable comments!

Yours sincerely

Hou

Reviewer 2 Report

Comments and Suggestions for Authors

I enjoyed reading this manuscript and have few criticisms.

There is a Word copy of your ms attached, with marginal comments and suggestions for different ways of wording things in the text. The current release of Word has eliminated the marginal markings that tell you where there is an emendation in the text, and the comments are no longer numbered. Presumably these features in older releases helped nobody but the user and so they were eliminated, but it means that you will have to look carefully at the text, as changes in punctuation may be difficult to spot. I would have marked up the PDF of your ms, but I find Adobe even worse to work with.

The written English is of a high quality, and I had little trouble in following the text. This applies to the organization of the text as well. Emendations that I have inserted into the text are generally suggestions of ways a native English speaker would have put things – I hope that you find them useful.

The statistics that you used were apposite, and I thought that your analysis was thorough and appropriate. Your discussion was thorough and cautious.

Most of what I have to say is in the annotations on this copy, and I will only discuss a few items here at greater length, or for emphasis; these are also referenced in the annotated copy of the ms accompanying this review:

-       Try to be consistent in referring to all of your home range size estimates as “estimates”. I have gone through the text and tried to change this when it was not done but probably missed a lot of instances.

-       Make sure that any R package used is denoted as an R package the first time that it is mentioned in the text. Usually, the name of the package is italicized when mentioned in a publication.

-       In Figures 1, 3, and 5, it is difficult or impossible to read the text – it was blurred, even when I magnified the figures. This detracts from their usefulness.

-       Is there some reason for not including the data points on the two plots comprising Figure 6?

-       The area in which you were working is not generally well-known. A short description of the topography and the predominant terrestrial vegetation, and the degree of human modification, would be helpful to foreign readers. A couple of sentences, with references to relevant authorities, would suffice.

-       I have asked for more detail on the handling of the turtles, particularly with regard to the method used to affix the transmitters to their shells. There is no question that the way you attached the transmitters was really the only way that you could have done it, but I asked about any means that you might have taken to reduce the trauma to the turtles of having holes drilled through their shells, and if you took steps to reduce the possibility of infection. The way they’re treated could have affected the way that turtles behaved after release – this might explain why one female swam 55 km upriver after release (incidentally, this particular instance of long-distance movement would make a nice natural history note for Herpetological Review – data like these are useful but seldom published). You stated that you gave turtles a week to recover in the wild before the commencement of data collection, but did you see any other instances of turtles, after their release, immediately moving some distance away from their initial capture point? This would indicate that their behaviour had been affected by the transmitter securing process.

-       Did you examine the holes drilled in the turtles’ carapaces upon the end of the study for evidence of infection etc.? An explicit statement of their condition upon final release is necessary.

-       Did you have to submit an animal handling protocol to your parent institute? Many journals require documentation of such protocols. I’ve raised this point with the editors of Animals – they will tell you if this is the case.

-       You gave the incidence of fishermen’s accidental capture of turtles in Appendix A, but, given the critical conservation status of this species in this part of its range, some comment on this in the text would be of value. How often do turtles from this population wind up being taken for the market? This study contains information relevant to conservation measures for this species and any data adding to that would increase its usefulness in this regard, even though it is tangential to the main theme of the paper. As you point out yourselves, there is little available in the literature on this species.

-       You mention that it was difficult to amass a sample of suitably-sized turtles, and you have more subadults than adults in your sample. This suggests that relatively few turtles are surviving to breeding age in this population, which has implications for its continued existence. Again, given its conservation status, this would be worth remarking upon (or dismissing, if I’ve interpreted this incorrectly).

-       It sounds as though Pelodiscus sinensis is found in this area as well, from your text (you state that local authorities don’t distinguish between this species and the Amur softshell turtle). Please make it clear that this is not the case.

There is a recent paper on repairing a puncture wound in a softshell turtle’s carapace – Pelodiscus maackii, as it happens – in a veterinary journal - Ha, M., D. N. Lee, S. Ahmed, J. Han, and S.-C. Yeon (2022) - Successful carapace puncture wound repair with polymethyl methacrylate (PMMA) in an Amur Softshell Turtle (Pelodiscus maackii). J Vet Clin 39:185-191.  https://doi.org/10.17555/jvc.2022.39.4.185 This paper may suggest means of patching up turtle shells at the end of the field season, although it describes clinical procedures that might not be possible to follow completely in the field.

I hope that all of this is helpful. This is good work and, given the apparent status of most chelonian species in East Asia, is timely and fills a gap in our knowledge of the basic ecology of the Amur softshell turtle. I trust that you will be able to continue with your research on these turtles in this part of their range and expand our understanding of this poorly-known species, and look forward to this particular contribution being published.

Comments on the Quality of English Language

The English was of good quality and I had no trouble following it. I have made some suggestions for re-phrasing or alternate words in the annotated copy of the ms accompanying this review, but for the most part these are suggestions of how a native English speaker would have put things.

Reviewer 3 Report

Comments and Suggestions for Authors

This manuscript reports on a small study using radiotransmitters to determine the home range and movements of an endangered turtle. The types of data collected in this manuscript are potentially useful for managing and conserving endangered or threatened species of freshwater turtles. The data are limited in extent and number of tracked turtles but even so the results are of interest. Below I provide some feedback I hope helps to improve the manuscript prior to publication.

line 88: Can you provide a citation or details about the specially designed traps?

lines 91-92: How did the fisherpeople collect the turtles?

line 113: How often did you track the turtles per day? Was it a single time or multiple times? If it was a single time when was it done?

            In addition, how many times did you locate each turtle? Provide the mean and range of the number of times each turtle was located.

lines 120-121: Where and when did you measure the water temperature? This is important to figure out how reasonable it is to use the measured temperatures to describe the behavior of the turtles.

line 215: How did you determine “complete” home range? You need to make your criteria explicit. I assume you mean that the home range size reached an asymptote but you need to make that clear for your reader.

line 293: What does “The rest interaction terms” mean?

line 340: I don’t think you mean “radiograph”. A radiograph is an x-ray image. I think you mean radiolocation.

Comments on the Quality of English Language

The English is pretty good but another quick review would be helpful.
